# Spatio-Temporal Patterns and Driving Factors of Vegetation Change in the Pan-Third Pole Region

**Xuyan Yang** [1,2]**, Qinke Yang** [1,2,*] **and Miaomiao Yang** [1,2]

1    Shaanxi Key Laboratory of Earth Surface System and Environmental Carrying Capacity, College of Urban and Environmental Sciences, Northwest University, Xi'an 710127, China
2    Key Laboratory of National Forestry Administration on Ecological Hydrology and Disaster Prevention in Arid Regions, Northwest University, Xi'an 710127, China
*    Correspondence: qkyang@nwu.edu.cn; Tel.: +86-13609259298

**Abstract:** The Pan-Third Pole (PTP) region, one of the areas with the most intense global warming, has seen substantial changes in vegetation cover. Based on the GIMMS NDVI3g and meteorological dataset from 1982 to 2015, this study evaluated the spatio-temporal variation in fractional vegetation coverage (FVC) by using linear regression analysis, standard deviation, correlation coefficient, and multiple linear regression residuals to explore its response mechanism to climate change and human activities. The findings showed that: (1) the FVC was progressively improved, with a linear trend of $0.003 \bullet 10a^{-1}$. (2) The largest proportion of the contribution to FVC change was found in the unchanged area (39.29%), followed by the obvious improvement (23.83%) and the mild improvement area (13.53%). (3) The impact of both climate change and human activities is dual in FVC changes, and human activities are increasing. (4) The FVC was positively correlated with temperature and precipitation, with a stronger correlation with temperature, and the climate trend was warm and humid. The findings of the study serve to understand the impacts of climate change and human activities on the dynamic changes in the FVC and provide a scientific foundation for ecological conservation and sustainable economic development in the PTP region.

**Keywords:** FVC; spatial-temporal vegetation patterns; climate change; human activities; Pan-Third Pole region

## 1. Introduction

Vegetation is a natural link between soil, atmosphere, and water. It plays a crucial role in energy exchange, water cycling, and biogeochemical cycling on the land surface and is sensitive to the stressful effects of human activities [1,2]. The quantitative assessment of the Intergovernmental Panel on Climate Change (IPCC) Fifth Assessment Report states that, "human activities are likely to have contributed to more than half of the increase in global mean surface temperature (GMST) from 1951 to 2010" [3]. The recently released IPCC AR6 Working Group I report demonstrated that human activities have led to the warming of the atmosphere, oceans, and land since the industrial revolution [4,5]. In the middle-high latitudes of the Northern Hemisphere, previous studies determined that global warming is the primary driving force of prolonged vegetation growth periods [6–9]. Climate change directly influences vegetation growth by changing vegetation metabolism [10]. Vegetation change is extremely sensitive to climate change, exhibiting dynamic and evolutionary traits, and is employed as an "indicator" of global climate change research [11–13]. The fractional vegetation coverage (FVC), which is typically defined as the vertical projection of vegetation (including leaves, stems, and branches) on the ground as a percentage of the total statistical area, represents the density of vegetation and the size of the photosynthetic area of plants [14]. It is an enhancement to the normalized difference vegetation index (NDVI), which partially solves the problem of NDVI easily saturating vegetation with high coverage and makes it difficult to distinguish vegetation with low coverage. It is widely

used in remote sensing-based ecological and environmental change monitoring [15,16]. The Pan-Third Pole (PTP) region is experiencing rapid global warming [17]. The Third Pole, with the Tibetan Plateau as the core, is the region with the strongest global warming and the greatest uncertainty about future global climate change impacts [18]. Furthermore, the projected warming of some areas of the PTP region will exceed the 2 °C goal set by the Paris Agreement of the United Nations Framework Convention on Climate Change [17]. There is great uncertainty about the serious consequences of such drastic climate change on the ecology and on human activities in the PTP region [17]. The PTP region is the core area of "the Belt and Road region", and is also the most fragile ecological environment and the region with the strongest impact of human activities on earth, which is important for human existence and sustainable development [19]. The vegetation in the PTP region is sensitive to climate change and human activities [20]. Climate change is the internal factor driving vegetation change, whereas human activity is an external factor [21–24]. Under the influence of global climate change and human activities, it is crucial to investigate the spatial and temporal characteristics of vegetation cover in the PTP region and to quantify the correlation between human activities and different climate variables and vegetation cover to understand the environmental conditions, environmental protection, and environmental restoration in the PTP region.

The long time series of NDVI data is useful tools for researching the history of vegetation, monitoring current conditions, and expressing concern about its future [25]. The Global Inventory Monitoring and Modeling System (GIMMS) NDVI3g dataset is the NDVI data with the longest time-series and the broadest coverage, which has obvious advantages in reflecting the dynamic changes in vegetation and is one of the best datasets to describe the dynamic changes in the growth of vegetation [26–28]. Currently, many studies have been using this NDVI long-term archive to detect changes and trends from regional level to global level. In particular, many papers have focused on the core area of the Third Pole region—the Tibetan plateau [29–32]—and the NDVI showed an overall increasing trend. The vegetation cover shows a trend of greening in the middle and high latitudes of the Northern Hemisphere [33,34]. In the Silk Road Economic Belt region, greening and browning of the vegetation coexisted, with the shift occurring in 2000 [35]. In Central Asia, a general increasing trend was found in the NDVI [36–38], but Hao et al. (2020) found that the vegetation greening and browning initially coexisted until reaching a turning point in 1994, after which browning dominated [39]. Furthermore, Liu et al. (2021) found that High Mountain Asia (HMA) has generally followed a "warm-wet" trend, but the vegetation after 1998 has been browning; the main reason for the browning is the dual effects of warming and precipitation changes [40]. In northern Eurasia, there was a trend of greening to browning in vegetation cover, but a general trend of greening [41,42]. In addition, the vegetation cover in Inner Mongolia and Mongolia is severely degraded due to overgrazing [43,44]. Additional to the above hotspots, Rustanto et al. (2022) studied the correlation between NDVI and aerosol optical depth (AOD) in the Middle East region, which confirms the significant effect of vegetation cover as having an essential role in dust storm fluctuations [45]. Even detecting long-term trends in NDVI on a global scale, the results show a coexistence of positive and negative trends [46–49]. The existing research results show that there are variations in the trends in NDVI across different regions, differences in the influencing factors of NDVI changes in vegetation in different regions, and that the main influencing factors also differ across time periods. Small regional studies lack general applicability, whereas large regional studies obfuscate regional peculiarities and fail to provide a clear explanation of the evolutionary characteristics and influencing factors of vegetation in the Pan-Third Pole region. This study analyzed the spatio-temporal changes in the FVC and focused on interactions of climate change and human activities in the process of vegetation growth using GIMMS NDVI3g and meteorological dataset from 1982 to 2015 with linear regression analysis, standard deviation, multiple regression residuals and other methods. The first aim was to understand the driving mechanism of spatio-temporal changes in FVC in the PTP region with a fragile ecological environment

and strong human-land interaction. The second aim was to identify the main control areas of climate change and human activities to provide a scientific basis and decision support for the conservation of vegetation resources and the mechanisms of vegetation-climate system interaction and feedback.

## 2. Materials and Methods

### 2.1. Study Area

The PTP is in the hinterland of Asia and Europe, with the Tibetan Plateau as the majority of the Third Pole extending westward and including, the Tibetan Plateau, Pamirs Plateau, Hindu Kush, Tian Shan, Iranian Plateau, Caucasus, and Carpathians [17]. The PTP region hosts a substantial part of the "Silk Road Economic Belt" and consists of over 20 countries and regions with an area of 20 million square kilometers and a population of over 3 billion [17]. The terrain of the PTP region is complex and dominated by mountains, hills, and plains, extending from the Third Pole and expanding to the west and north. The massive mountain system that runs east to west generates the skeleton of the PTP landscape. The climate type is complicated, dominated by a normal monsoon-arid climate system, and the substratum is intricate, diverse, and vulnerable [50]. The temperature varies by up to 30 °C from north to south. Precipitation is affected by the monsoon system and annual precipitation varies significantly by region, with a maximum of 4800 mm in the southern Tibetan Plateau, South Asia, and central Europe, and minimum of less than 500 mm in the Middle East and Central Asia [50]. Forest ecosystem types predominate in Southeast Asia, Russia, and Europe; grassland and desert ecosystem types in Central Asia; farmland ecosystem types in South Asia; and farmland, forest, grassland, and desert ecosystem types are roughly proportional in East Asia [51]. Due to regional differences in climate change, the study area was divided into seven subregions (Figure 1) based on physical geography, socioeconomic development characteristics, and the degree of close exchange and co-operation with China to better highlight the spatial and temporal evolution characteristics of the FVC in different regions: East Asia (EAS), South Asia (SAS), Southeast Asia (SEA), Central Asia (CAS), West Asia (WAS), Central and Eastern Europe (excluding Russia) (CEU), and Russia (RUS).

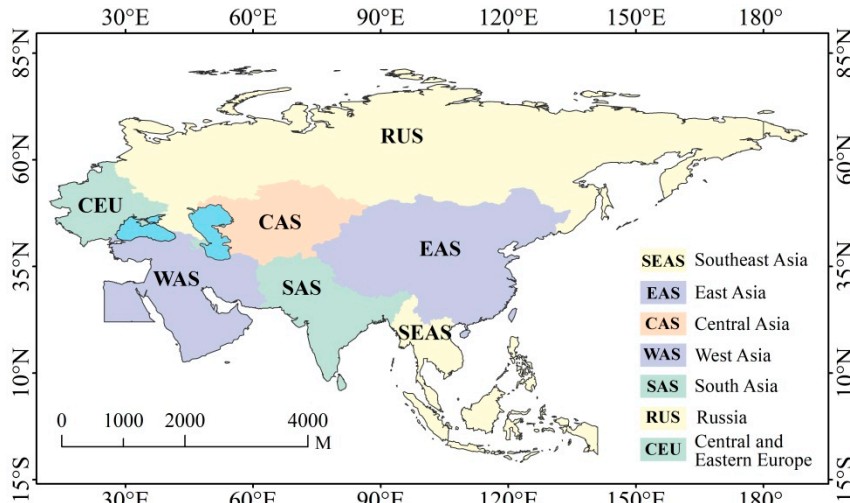

**Figure 1.** Subregions map in the Pan-Third Pole region.

### 2.2. Materials

NDVI data were obtained from the GIMMS NDVI3g dataset with a spatial resolution of 8 km and temporal resolution of 15 days (http://ecocast.arc.nasa.gov/pub/gimms/, accessed on 19 July 2022) (January 1982 to October 2015). During the synthesis and calculation process, the GIMMS dataset was pre-processed using radiometric correction, geometric correction, atmospheric correction, and co-ordinate conversion to ensure the

quality of the data was guaranteed [36]. Because the maximum NDVI value composite (MVC) (a maximum daily NDVI value every 15 days) minimizes atmospheric effects, scan angle effects, cloud contamination, and solar zenith angle effects [52], we used the largest 15-day MVC for a month to produce the monthly NDVI dataset. Land cover data were obtained from the MODIS standard land cover product MCD12Q1, with an International Geosphere–Biosphere Programme (IGBP) classification scheme and 8 km spatial resolution. Meteorological data (temperature and precipitation) with a spatial resolution of 5 km were obtained from Google Earth Engine (GEE) (http://earthengine.google.com, accessed on 19 July 2022) (January 1982 to October 2015) using the "TerraClimate: Monthly Climate and Climate Water Balance for Global Terrestrail Surface, University of Idaho" dataset. The spatial resolution was resampled to 8 km and projected as a cylindrical equal area.

### 2.3. Methods

In this paper, linear regression analysis, standard deviation, multiple regression residuals, and partial correlation analysis were used to analyze the spatio-temporal evolution characteristics and driving factors of vegetation change in PTP region. The workflow of this study is shown in Figure 2:

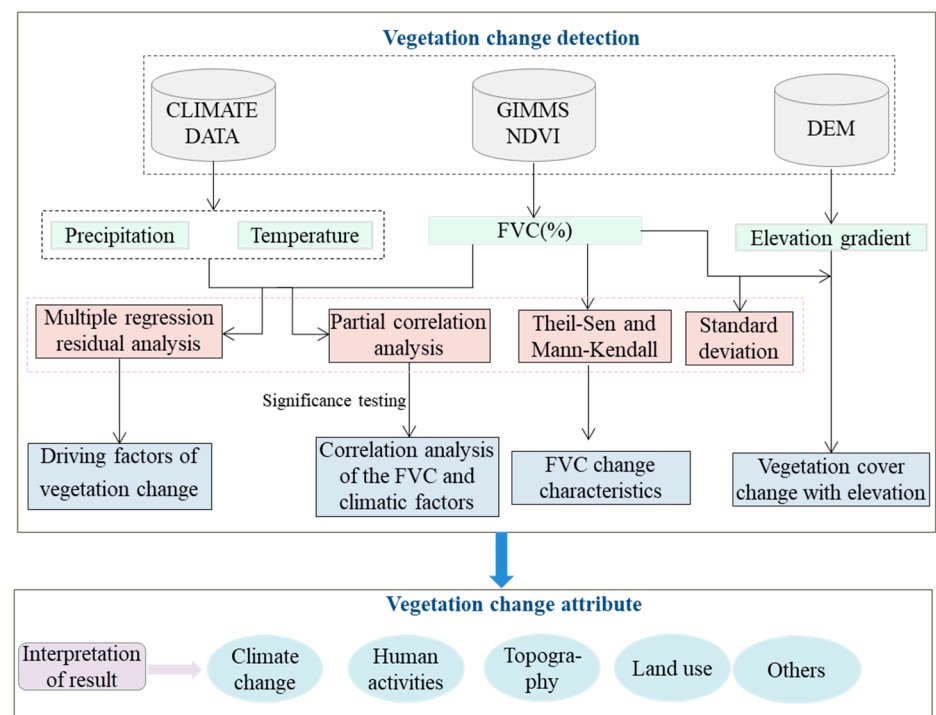

**Figure 2.** Map showing flowchart of this study.

### 2.3.1. Dimidiate Pixel Model

The *FVC* can effectively reduce the uncertainty caused by the spectral characteristics of the unvegetated regions and improve the accuracy of the analysis and has been widely used in previous studies [36,53,54]. Therefore, the *FVC* of the image is equal to the difference between the vegetation index and bare ground vegetation index [55]. The calculation formula is as follows:

$$FVC = (NDVI - NDVI_{soil})/(NDVI_{veg} - NDVI_{soil}) \tag{1}$$

where *FVC* is the fractional vegetation cover (%), *NDVI* is the normalized vegetation index value of the image element, $NDVI_{soil}$ ($NDVI_{min}$) is the *NDVI* value of pure soil pixels, and $NDVI_{veg}$ ($NDVI_{max}$) is the *NDVI* value of pure vegetation pixels. Because of the inherent noise in the image, the extreme *NDVI* values are not always $NDVI_{max}$ and

$NDVI_{min}$, hence, the values are decided by the image scale and image quality. In the absence of measured data, the $NDVI_{max}$ and $NDVI_{min}$ values are considered as the maximum and minimum values within the confidence interval for a certain confidence level of the image. In this study, $NDVI_{max}$ is defined as $NDVI$ values at a 95% cumulative frequency in forest, and $NDVI_{min}$ is defined as $NDVI$ values at a 5% cumulative frequency in bare land and sparse vegetation.

### 2.3.2. Trend Analysis

In this study, the spatial and temporal evolution trends of vegetation cover in the PTP region were analyzed using one-dimensional linear regression analysis from 1982 to 2015 and the stability of the $FVC$ in the PTP region was expressed as standard deviation (SD).

(1)    Theil–Sen median trend analysis and Mann–Kendall significance test

Theil–Sen median trend analysis is a robust non-parametric statistical trend method that can be used to reflect the $FVC$ trend [56,57]. In this study, the Theil–Sen median trend analysis was used to simulate the spatial and temporal trends of vegetation cover in the PTP from 1982 to 2015 and a raster-by-raster simulation of the trends of the $FVC$ in the study area. The calculation formulae are as follows:

$$Slope_{FVC} = Median\left(\frac{FVC_j - FVC_i}{j - i}\right) \tag{2}$$

$$1982 \leq i \leq j \leq 2015 \tag{3}$$

where $S_{FVC}$ is the median slope of n(n−1)/2 data combinations, and $FVC_i$ and $FVC_j$ are the average image element values for years $i$ and $j$ ($1982 \leq i \leq j \leq 2015$), respectively. When $Slope_{FVC} > 0$, the $FVC$ tends to increase; otherwise, the $FVC$ tends to decrease.

Mann–Kendall is a non-parametric test that may be used to determine if time-series data are trending upward or downward because it does not require the sample to follow a particular distribution and is unaffected by a few abnormalities [58,59]. The calculation formula is as follows:

Setting $\{FVC_i\}$, $i$ = 1982, 1983, . . . , 2015, define the Z-statistic as:

$$Z = \begin{cases} \frac{s-1}{\sqrt{var(S)}}, S > 0 \\ 0, S = 0 \\ \frac{s+1}{\sqrt{var(S)}}, S < 0 \end{cases} \tag{4}$$

$$S = \sum_{j=1}^{n-1} \sum_{i=j+1}^{n} sgn(FVC_j - FVC_i) \tag{5}$$

$$sgn(FVC_j - FVC_i) = \begin{cases} 1, FVC_j - FVC_i > 0 \\ 0, FVC_j - FVC_i = 0 \\ -1, FVC_j - FVC_i < 0 \end{cases} \tag{6}$$

$$var(S) = \frac{n(n-1)(2n+5)}{18} \tag{7}$$

where $n$ is the length of the year and *sgn* is the sign function. In this study, the significance test of $FVC$ change trends was judged by $a$ = 0.05. The trend of vegetation cover change in the study area over 34 years was classified into five classes by combining $S_{FVC}$ and $Z$ values (Table 1).

**Table 1.** The classification standard for *FVC* change trends.

| Theil-Sen | MK Significance Test | Change Levels | Theil-Sen | MK Significance Test | Change Levels |
|---|---|---|---|---|---|
| *Slope* $\leq -0.0005$ | $Z \leq -1.96$ | Serious degradation | *Slope* $\geq 0.005$ | $-1.96 < Z < 19.6$ | Mild improvement |
| *Slope* $\leq -0.0005$ | $-1.96 < Z < 19.6$ | Slight degradation | *Slope* $\geq 0.005$ | $Z > 1.96$ | Obvious improvement |
| $-0.0005 < Slope < 0.0005$ | $-1.96 < Z < 19.6$ | Unchange | | | |

(2)  Standard deviation (SD)

SD is a commonly employed measure of variation and diversity that demonstrates how much variation or dispersion exists around the mean (mean or expected value) [60]. A low standard deviation indicates that the *FVC* values of each image element tend to be very close to the mean, whereas a high standard deviation indicates that the *FVC* of each image element is farther from the mean.

$$S_i = \sqrt{\frac{1}{n} \sum_{i=1}^{n} \left( FVC_i - \overline{FVC} \right)^2} \tag{8}$$

where $S_i$ is the standard deviation (SD), $i$ is the study year, $FVC_i$ is the *FVC* value in year $i$, and $\overline{FVC}$ is the multi-year average vegetation cover for the study period. The larger the SD value, the greater the variation indicated, and vice versa. The Natural Breaks (Jenks) method was used to classify the SD into five categories: High fluctuation ($S_i > 0.185$), higher fluctuation ($0.052 < S_i < 0.185$), moderate fluctuation ($0.032 < S_i < 0.052$), lower fluctuation ($0.015 < S_i < 0.032$), and low fluctuation ($0 < S_i < 0.015$).

2.3.3. Multiple Regression Residual Analysis

Multiple regression residual analysis is a prevalent technique for analyzing the impacts and relative contributions of human activities and climate change on plant cover change [61,62]. The method had three steps: (1) based on the *FVC* in the growing season and the time-series data of annual mean temperature and annual precipitation, a binary linear regression model was established with *FVC* as the dependent variable and temperature and precipitation as the independent variables, and the parameters in the model were calculated. (2) Based on the temperature and precipitation data and the parameters of the regression model, the predicted value of the *FVC* ($FVC_{CC}$) was calculated, which represents the influence of climate factors on vegetation FVC. (3) The difference between the observed value of the *FVC* ($FVC_{obs}$) and the predicted value of the FVC ($FVC_{CC}$), that is, the *FVC* residual ($FVC_{HA}$), was calculated to represent the influence of human activities on the *FVC*. The specific calculation formula was as follows:

$$FVC_{CC} = a \times T + b \times P + c \tag{9}$$

$$FVC_{HA} = FVC_{obs} - FVC_{CC} \tag{10}$$

where $FVC_{obs}$ and $FVC_{CC}$ represent the observed value of the *FVC* based on remote sensing images and the predicted value of the *FVC* based on regression models (dimensionless), respectively; *a*, *b*, and *c* are model parameters; T and P is the average temperature and accumulated precipitation of the growing season in °C and mm, respectively; and $FVC_{HA}$ is the residual. $FVC_{HA} > 0$ indicates a positive impact of human activities, $FVC_{HA} < 0$ indicates a negative impact of human activities, and $FVC_{HA} = 0$ indicates a relatively weak impact of human activities.

### 2.3.4. Partial Correlation Analysis

The geographic system is a complex multifactor system, especially in the system composed of many factors, and change in one variable will inevitably impact another variable. When two variables are associated with a third variable, partial correlation analysis disregards the influence of the third variable and focuses on the correlation between the first two [63]. The partial correlation coefficients of the *FVC* with precipitation and temperature were calculated as follows:

$$r_{xy} = \frac{\sum_{i=1}^{n}[(x_i - \bar{x})(y_i - \bar{y})]}{\sqrt{\sum_{i=1}^{n}(x_i - x)^2 \sum_{i=1}^{n}(y_i - y)^2}} \tag{11}$$

where $r_{xy}$ is the partial correlation coefficient between the two variables $x$ and $y$, whose value ranges from $-1$ to 1; $x_i$ and $y_i$ are the *FVC* in the $i$th year and mean temperature or precipitation, respectively; $x$ is the mean *FVC* from 1982 to 2015; and $\bar{y}$ is the mean of the temperature or precipitation in the growing season. The partial correlation coefficients passed the significance level determined using the $t$-test.

### 2.3.5. Driving Factors of *FVC* Change

The linear trend rates of $FVC_{CC}$ and $FVC_{HA}$ in the PTP region from 1982 to 2015 were calculated using Equation (2), showing the trends in *FVC* changes due to climate change and human activities, respectively. When *Slope* > 0, climatic change or human activities had a positive effect on vegetation growth; conversely, when *Slope* < 0, they had a suppressive effect. To properly analyze the impacts of climate change and human activities on vegetation growth status, Table 2 identifies the main drivers of *FVC* changes throughout the PTP growing season [64].

**Table 2.** Determination criteria for driving forces of *FVC* change.

| Slope(FVC$_{obs}$) | Driving Forces | Criteria for Classifying Driving Factors | |
| --- | --- | --- | --- |
| | | Slope(FVC$_{CC}$) | Slope(FVC$_{HA}$) |
| | CC & HA | >0 | >0 |
| >0 | CC | >0 | <0 |
| | HA | <0 | >0 |
| | CC & HA | <0 | <0 |
| <0 | CC | <0 | >0 |
| | HA | >0 | <0 |

Note: *Slope(FVC$_{obs}$)*, *Slope(FVC$_{CC}$)*, and *Slope(FVC$_{HA}$)* represent the growth season *FVC* observations based on remote sensing data, growing season *FVC* predictions based on multivariate residual analysis, and growing season *FVC* residuals, respectively. *CC* and *HA* represent the impact of climate change and human activities, respectively.

## 3. Results

### 3.1. Spatial and Temporal Evolution Characteristics of the FVC

#### 3.1.1. Changes in Overall Characteristics

The statistical chart of the annual average value of the *FVC* in the PTP region over 34 years showed a fluctuating and slowly increasing trend (Figure 3a), with a value range of 43.79–47.59% and multi-year average of 45.65%. The *FVC* reached its highest value in 1994 and lowest value in 1985 and fluctuated around 45.5% until the year 2000 when it stabilized. From 1982 to 2015, the average *FVC* trend rate in the PTP region was $0.003 \bullet 10a^{-1}$, showing greater growth in the *FVC* in the studied area. At the same time, a phase change in the rising trend of *FVC* is observable, with 1982 to 1994 being a rapid rising period, with a growth rate of $0.0018 \bullet a^{-1}$; from 1994 to 2003, the vegetation browns, with a decline rate of $-0.0008 \bullet a^{-1}$; and after 2003, the vegetation gradually greens, with a growth rate of $0.0008 \bullet a^{-1}$.

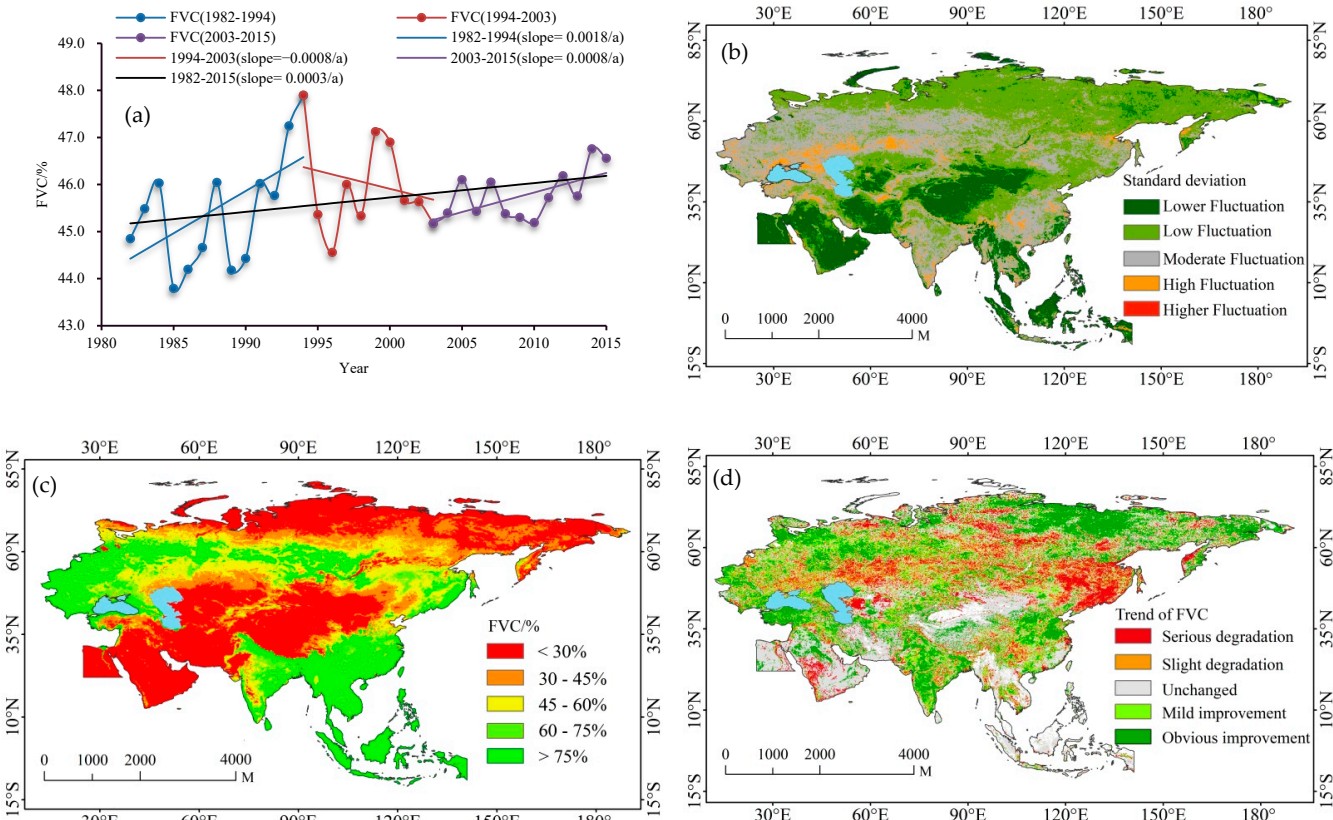

**Figure 3.** Inter-annual variations in annual mean *FVC* in the Pan-Third Pole region during 1982–2015 (**a**); inter-annual variations standard deviation of *FVC* (**b**); spatial pattern in mean *FVC* (**c**); and change trend of *FVC* based on pixels (**d**).

During the PTP region growing season from 1982 to 2015, the range of the *FVC* standard deviation values was between 0 and 0.49, with an average SD value of 0.03, which is near to a normal distribution. The spatial stability pattern shows that there were significant regional variances in *FVC* stability (Figure 3b). The proportions in each classification were ranked as low fluctuation (37.41%) > moderate fluctuation (32.73%) > lower fluctuation (23.56%) > higher fluctuation (6.24%) > high fluctuation (0.06%). The low fluctuation region was found primarily in the Central Siberian Plateau, east Siberian Mountains, and northern Central Asia; the moderate fluctuation region was distributed in the primarily southern West Siberian Plain, Eastern European Plain, and middle and lower Yangtze River Plain in China; the lower fluctuation region was mostly in the Qinghai–Tibet Plateau, Iranian Plateau, West Asia, and Southeast Asia; and the higher fluctuation and high fluctuation region was more scattered and lacked agglomeration.

Figure 3c shows the spatial pattern of multi-year average *FVC* in the PTP region, with a mean *FVC* of 45.58%, which is a medium-low cover level. The high-value regions were mostly in the middle-lower Yangtze River Plain, Eastern European Plain, and southern Siberian Plain, whereas the low-value regions were primarily in western Mongolia, Qinghai–Tibet Plateau, Central Asia, West Asia, and Eastern and Northern Russia (excluding Russia). Overall, the lack of light and heat energy caused by the perennial low temperature in the high-latitude regions is not suitable for the growth of vegetation and the *FVC* value was lower. The mixed forests and evergreen needle-leaf forests in the middle-latitude regions occupy a larger area and, thus, the *FVC* value was moderate in these regions. Low-latitude regions are dominated by evergreen needle-leaf forests and evergreen broad-leaf forests in the east and, thus, the *FVC* value was higher.

A statistical multi-year trend rate (*Slope*) by pixel for *FVC* in the PTP region shows that average trend rate of 0.003$\bullet$10a$^{-1}$ over 34 years (Figure 3d). The spatial pattern

evolution characteristics were as follows: (1) the proportion of serious degradation was 12.19% and the proportion of slightly degradation was 11.16%, with concentrations in the middle Siberian Plateau, Northeast Plain, western Saudi Arabia, Kazakhstan, and eastern Aral Sea; (2) the proportion of unchanged was 39.29%, primarily in northwestern China, northern Central Asia, western South Asia; (3) the proportion of obvious improvement was 39.29% and the proportion of mild improvement was 13.53%, mostly found in East Siberian highlands, Eastern European plains, India, and north-central China at high latitudes. Overall, there was high stability of the *FVC* in the study area over 34 years, with twice as many obvious improvements as serious degradation, a slow increase in the *FVC*, and gradual ecological improvements.

### 3.1.2. Trend Analysis of Subregions

The multi-year mean *FVC* in subregions was ranked as Southeast Asia (94.16%) > Central and Eastern Europe (excluding Russia) (78.38%) > South Asia (51.15%) > Russia (46.95%) > East Asia (40.88%) > Central Asia (24.91%) > West Asia (17.08%). The average trend rate (*Slope*) of the *FVC* in subregions was ranked as South Asia ($0.68 \bullet 10a^{-1}$) > Central and Eastern Europe (excluding Russia) ($0.47 \bullet 10a^{-1}$) > West Asia ($0.44 \bullet 10a^{-1}$) > East Asia ($0.18 \bullet 10a^{-1}$) > Russia ($0.17 \bullet 10a^{-1}$) > Central Asia ($0.12 \bullet 10a^{-1}$) > Southeast Asia ($0.01 \bullet 10a^{-1}$), with South Asia increasing the fastest and Southeast Asia increasing the slowest but still with an overall growing trend (Figure 4). The spatial-temporal variation in the *FVC* in subregions was statistically analyzed (Table 3). The largest proportion of obvious improvement was in South Asia (33.98%) and the smallest in Southeast Asia (8.19%). The largest proportion of serious degradation was in Central Asia (14.96%), while Central and Eastern Europe (excluding Russia) had the smallest proportion (6.32%). Southeast Asia had the highest vegetation stability with 67.22% and Central and Eastern Europe (excluding Russia) had the lowest vegetation stability with the smallest proportion (25.22%). Figure 3d shows the significant geographic variation in *FVC* trends within the subregions, and seven sample regions were selected for assessment based on land use types (Figure 5). Sample6 showed the most obvious greening trend with a slope of $0.015 \bullet 10a^{-1}$; followed by sample1, with a slope of $0.012 \bullet 10a^{-1}$. Sample4 showed the most obvious browning tendency (*Slope* $= -0.015 \bullet 10a^{-1}$), followed by the slope of sample2 is $-0.014 \bullet 10a^{-1}$ (Figure 6). The mean *FVC* was stable or slightly increasing in all subregions. However, the percentage of degraded areas may be about 30% in Central Asia, 28.87% in Russia, and 24.34% in East Asia. The browning of vegetation is a much more important issue that must be given attention. In general, the regions where the *FVC* tended to improve were in shrubs and mixed forests at middle and high latitudes, whereas the *FVC* in humid and semi-humid regions at relatively low elevations in Southeast Asia and Central Asia showed a degradation trend. Global warming may have improved vegetation growth conditions in high-altitude and cold regions, while the degradation of vegetation in regions with lower altitudes and better climatic conditions may be caused by human activities.

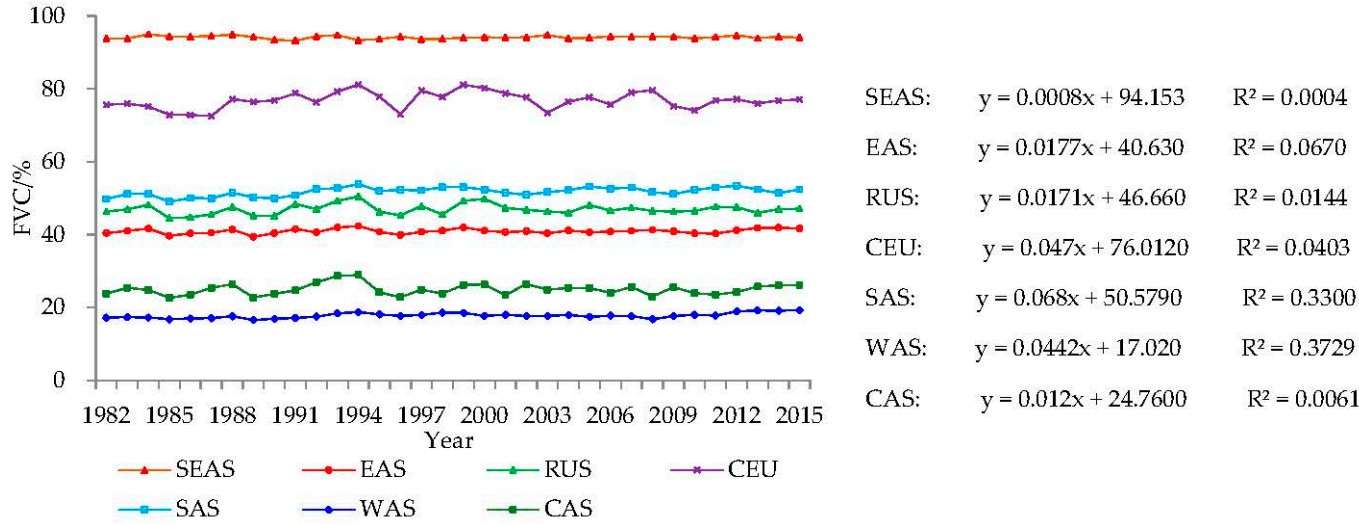

**Figure 4.** Inter-annual variations in annual mean *FVC* of subregions.

**Table 3.** Change trend statistics for *FVC* in subregions.

| Region | Serious Degradation (%) | Slight Degradation (%) | Unchanged (%) | Mild Improvement (%) | Obvious Improvement (%) |
|---|---|---|---|---|---|
| SEAS | 8.24 | 8.08 | 67.22 | 8.28 | 8.19 |
| EAS | 13.71 | 10.63 | 43.21 | 12.66 | 19.79 |
| RUS | 14.71 | 14.16 | 28.56 | 14.98 | 27.59 |
| CEU | 6.32 | 17.42 | 25.22 | 23.06 | 27.98 |
| SAS | 6.42 | 8.12 | 33.67 | 17.81 | 33.98 |
| WAS | 9.87 | 3.65 | 55.85 | 6.37 | 24.26 |
| CAS | 14.96 | 15.87 | 38.79 | 15.72 | 14.66 |

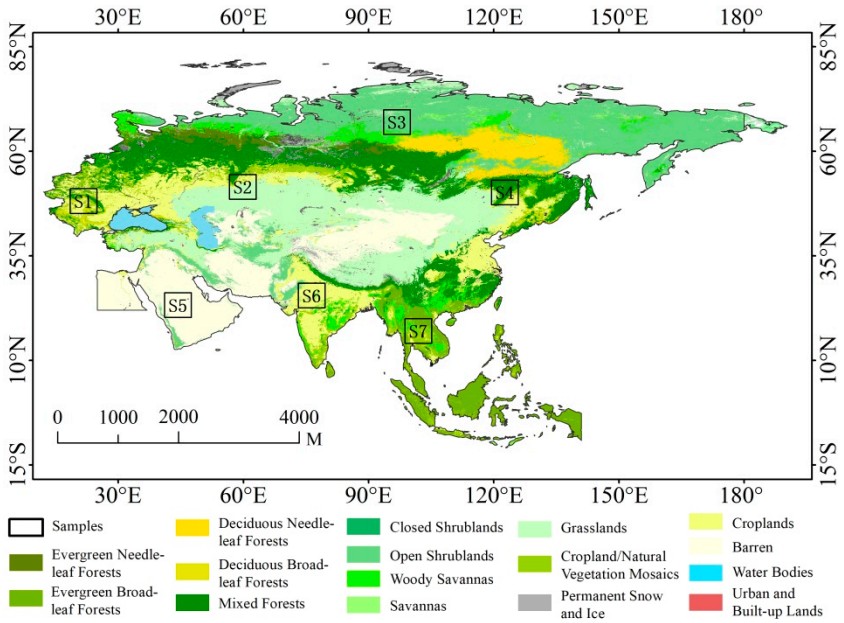

**Figure 5.** Samples and Land use map of Pan-Third Pole region based on IGBP.

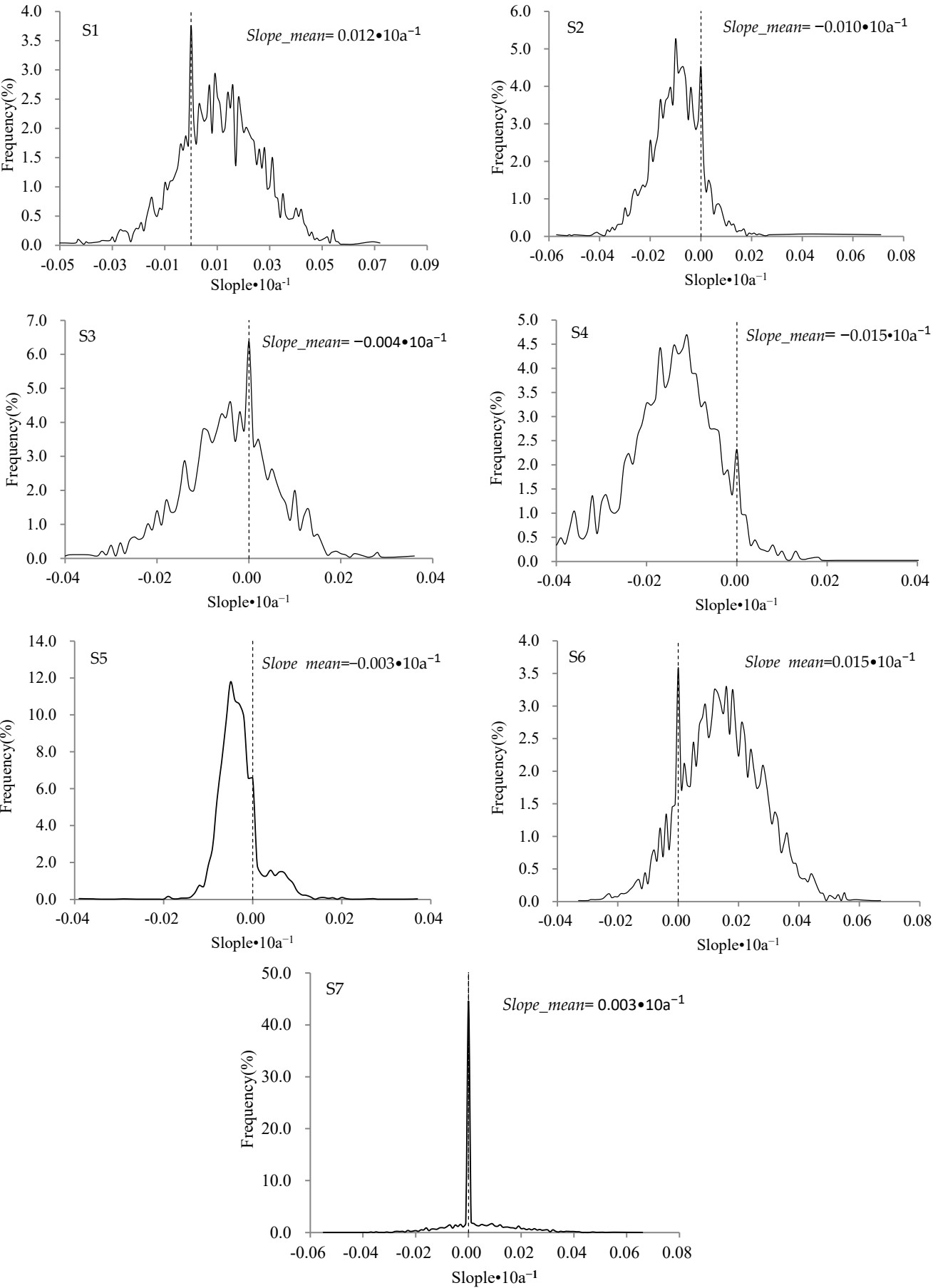

**Figure 6.** Histograms of *FVC* trend in seven samples (S1–S7 denote Sample1–7, respectively).

### 3.2. Analysis of the Driving Factors of the FVC

#### 3.2.1. Response of *FVC* Change to Climate Change

From 1982 to 2015, the geographic variability of $FVC_{CC}$ in the PTP region was significant, ranging from 0.078 to $0.25 \bullet 10a^{-1}$ and with a mean value of $0.0013 \bullet 10a^{-1}$ (Figure 7). Central and Eastern Europe (excluding Russia) experienced the greatest rate of increase in the *FVC* (*Slope* = $0.0061 \bullet 10a^{-1}$), followed by East Asia (*Slope* = $0.0014 \bullet 10a^{-1}$) and Central Asia (*Slope* = $0.00059 \bullet 10a^{-1}$), and only Southeast Asia exhibited a decreasing trend in $FVC_{CC}$ under the influence of climate change (*Slope* = $-0.00032 \bullet 10a^{-1}$). Statistical analysis showed that the proportion of climate change having a suppressive effect on the increase in vegetation cover was 42.41% and was found mostly in the northeastern Aral Sea, southern West Asia, central Russia, and Southeast Asia. The proportion of climate change contributing to the increase in the *FVC* was 57.59%, found mostly in northern Russia, Central and Eastern Europe, eastern China, and northern West Asia. The trend of $FVC_{CC}$ shows that, except Southeast Asia, all subregions were positively affected by climate change.

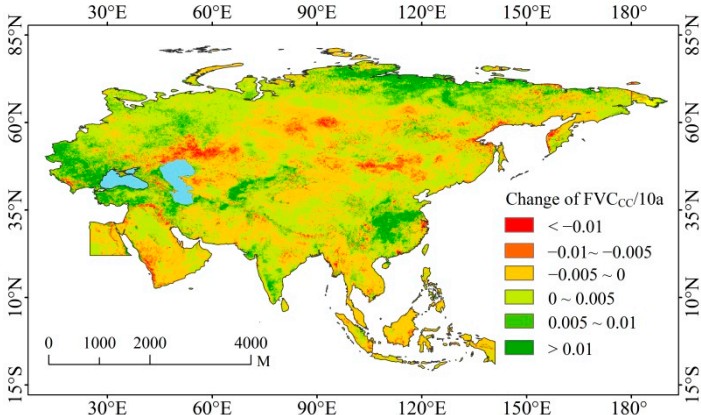

**Figure 7.** Change trends of $FVC_{CC}$ in the Pan-Third Pole region.

The $FVC_{CC}$ trends (*Slope*) were computed for the subregions (Figure 8), showing that climate change had the greatest suppression on the increase in vegetation in Southeast Asia, with the proportion of suppression as high as 71.38%. Climate change had the most obvious promoting effect on the increase in the *FVC* in Central and Eastern Europe (excluding Russia), with the percentage of promoting growth reaching 84.63%, and the suppressing and promoting effects of climate change on the increase in vegetation cover in South Asia, West Asia, and Central Asia were comparable. Climate change promoted the growth of vegetation at high latitudes but suppressed the growth of vegetation at low latitudes, while vegetation changes in mid latitudes responded insignificantly to climate change.

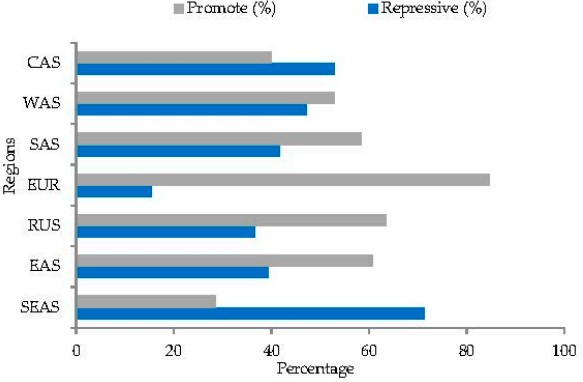

**Figure 8.** Impact of climate changes on $FVC_{CC}$ in subregions.

### 3.2.2. Response of the *FVC* Change to Human Activities

The $FVC_{HA}$ in the PTP region gradually increased from 1982 to 2015, ranging from $-0.14$ to $0.40 \bullet 10a^{-1}$ and with a mean value of $0.0011 \bullet 10a^{-1}$ (Figure 9). South Asia had the highest rate of $FVC_{HA}$ (*Slope* = $0.0068 \bullet 10a^{-1}$), followed by West Asia (*Slope* = $0.0024 \bullet 10a^{-1}$), and Southeast Asia had the slowest rate (*Slope* = $0.0002 \bullet 10a^{-1}$). Conversely, the $FVC_{HA}$ in Central and Eastern Europe (excluding Russia), East Asia, and Central Asia showed a decreasing trend as a result of human activities, with the greatest rate of decrease found in Central and Eastern Europe (excluding Russia) (*Slope* = $-0.0008 \bullet 10a^{-1}$), followed by Central Asia (*Slope* = $-0.0006 \bullet 10a^{-1}$) and East Asia (*Slope* = $-0.0003 \bullet 10a^{-1}$). Statistical analysis showed that the proportion of the area where human activities had a suppressing effect on the increase in the *FVC* was 47.11%, mainly in Southeast Asia, central Russia, and northeastern East Asia. These regions have more developed economies and higher population densities and economic growth, including engineering and construction activities, which has decreased the *FVC*. Due to human activities, the *FVC* has risen by 52.89%, primarily in eastern China, northern West Asia, southern South Asia, and northeastern Russia. In these regions, the implementation of ecological projects has a significant impact on ecological engineering and human activities that promote vegetation growth.

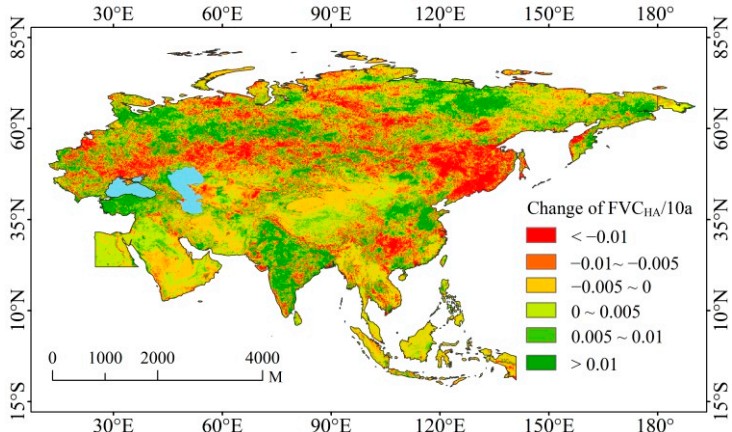

**Figure 9.** Change trends of $FVC_{HA}$ in the Pan-Third Pole region.

Statistics on the trends (*Slope*) of $FVC_{HA}$ in the subregions are shown in Figure 10, which reveals that human activities had the greatest suppressing effect on the increase in vegetation cover in Central Asia, with the proportion of suppression up to 56.26%. Human activities had the most obvious promoting effect on the increase in vegetation cover in South Asia, with the proportion of promoted growth up to 67.63%. Human activities had comparable suppressing and promoting effects on the vegetation cover in East Asia, Russia, and Central and Eastern Europe (excluding Russia). The suppressive and promoting effects of human activities on the *FVC* increase were comparable. Human activities had a promoting effect on vegetation restoration in the middle and low latitudes, while the suppressing effect on vegetation cover in arid and semi-arid areas at low latitudes and sparse shrubs at high latitudes was obvious. The positive driving and negative suppressing effects of human activities on vegetation growth co-exist, while the promoting effect was greater than the suppressing effect.

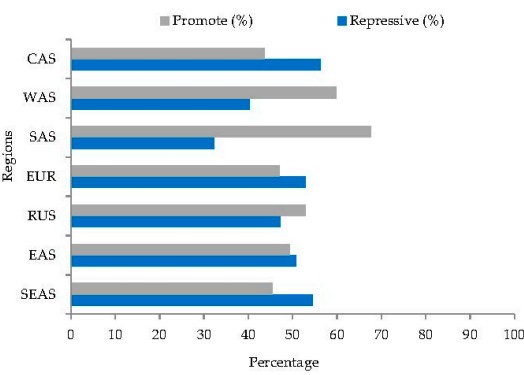

**Figure 10.** Impact of human activities on $FVC_{HA}$ in subregions.

### 3.2.3. Driving Factors of *FVC* Change

The effects of climate change and human activities on the *FVC* variability in the PTP region showed significant spatial variation (Figure 11) and the effects of the two factors on *FVC* variations in the same region were significantly different. Among the seven subregions, the trend rate (*Slope*) of *FVC* change was greater than 0. South Asia had the greatest increase in the *FVC* (*Slope* = 0.0079•10a$^{-1}$), followed by Central and Eastern Europe (excluding Russia) (*Slope* = 0.0061•10a$^{-1}$), whereas Central Asia had the slowest increase in the *FVC* (*Slope* = 0.0002•10a$^{-1}$). In general, (1) climate change contributes to the increase in the *FVC* in approximately 34.71% of regions, primarily in southeastern and western Russia, northern West Asia, eastern China, and India; the proportion of regions in which climate change suppressed the increase in the *FVC* was nearly 6.32%, primarily in northern Russia and northern South Asia and dispersed in other regions. The area where human activities promoted an increase in the *FVC* was nearly 8.58%, primarily in northern South Asia, northern East Asia, and the south-central region of Russia; (2) the area of 1.75% of the decreasing *FVC* in the entire PTP region was the result of the combined effect of climate and human activities; with climate change playing a dominant role in 5.93%, primarily in Southeast Asia. Human activities dominated in 12.51% of regions, mostly in the northeastern portion of East Asia and northwestern portion of Russia; (3) the 30.2% change in vegetation cover in the PTP region was unrelated to climate change or human activities and was mostly found in the north of Central Asia, Central Asia, the Qinghai–Tibet Plateau, and other areas with low vegetation cover and population density.

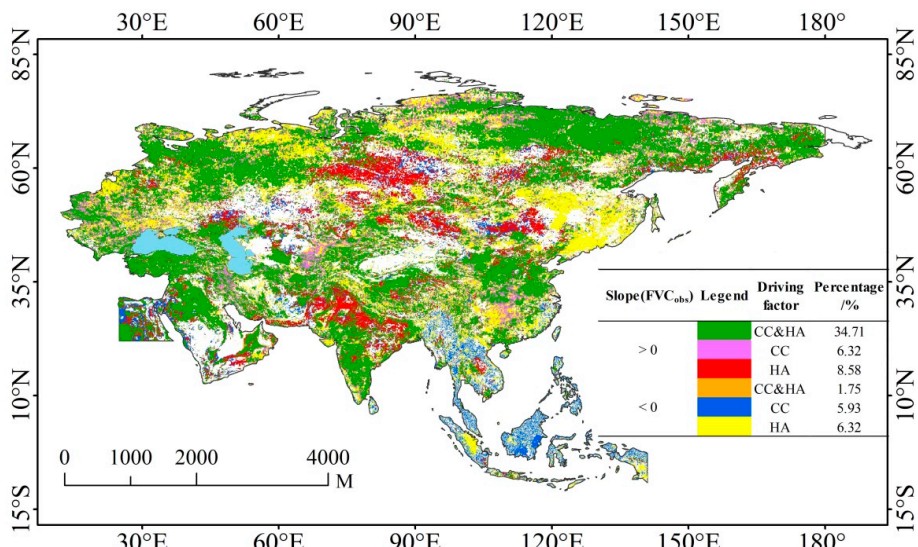

**Figure 11.** Spatial distribution of driving factors of *FVC* changes in the Pan-Third Pole region from 1982 to 2015 (CC and HA represent climate change and human activities, respectively).

According to the statistical results in Table 4, climate change and human activities were the primary driving forces of increasing *FVC* in South Asia, West Asia, and Russia; climate change was the primary driving force of increasing *FVC* in Central and Eastern Europe (excluding Russia), East Asia, and Central Asia; and human activities were the main driving force for increasing *FVC* in Southeast Asia. In the past 34 years, climate change and human activities contributed more to the increase in the *FVC* in the PTP region than they suppressed it, resulting in a gradual increase in the *FVC*.

**Table 4.** *FVC* trend rates and driving factors of subregions in the Pan-Third Pole region during 1982–2015 ($10a^{-1}$).

| Regions | *Slope (FVC_{obs})* | Effect on Vegetation Restoration | | Driving Factors |
|---|---|---|---|---|
| | | **CC** | **HA** | |
| SAS | 0.0078 | Promote | Promote | CC&HA |
| CEU | 0.0061 | Promote | Repressive | CC |
| WAS | 0.0033 | Promote | Promote | CC&HA |
| SEAS | 0.0004 | Repressive | Promote | HA |
| EAS | 0.0012 | Promote | Repressive | CC |
| CAS | 0.0002 | Promote | Repressive | CC |
| RUS | 0.0022 | Promote | Promote | CC&HA |

Note: *Slope (FVC_{obs})* represent the growing season *FVC* observations based on remote sensing data. CC and HA represent the impacts of climate change and human activities, respectively.

### 3.3. Correlation Analysis of the FVC and Climatic Factors

3.3.1. Characteristics of Inter-Annual Variability of Climate Factors

During the 34-year period, the mean temperature in the PTP region was 4.86 °C, with an average trend rate of 0.326 °C•$10a^{-1}$ and a generally slow increasing trend (Figure 12a), accompanied by a warming trend in the general climatic background. Since 1982, the highest temperature was 5.99 °C in 2007, which was 23% above the mean temperature, whereas the lowest temperature was 3.88 °C in 1987, which was 20% below the mean temperature. The annual mean temperature and trend of change in the study area varied significantly in space and time (Figure 13a,b), ranging from −21.01 to 30.28 °C and decreasing with latitude (the average altitude of the Qinghai–Tibet Plateau is above 4000 m, which causes the annual mean temperature to be lower) and temperature at high latitudes to gradually decrease with increasing longitude, showing general trends of decreasing from south to north. From southwest to northeast, the main tendency was downward. Additionally, there were significant regional differences in annual mean temperatures among the seven subregions, in the following order: Southeast Asia (24.08 °C) > South Asia (20.71 °C) > West Asia (20.11 °C) > Central and Eastern Europe (excluding Russia) (8.98 °C) > Central Asia (8.44 °C) > East Asia (5.56 °C) > Russia (−4.78 °C). The high-temperature regions were mainly in the lower latitudes of the Indian Peninsula and Arabian Peninsula, while temperature zones with low values were found in the Qinghai–Tibet Plateau and the majority of northern Mongolia and Siberia.

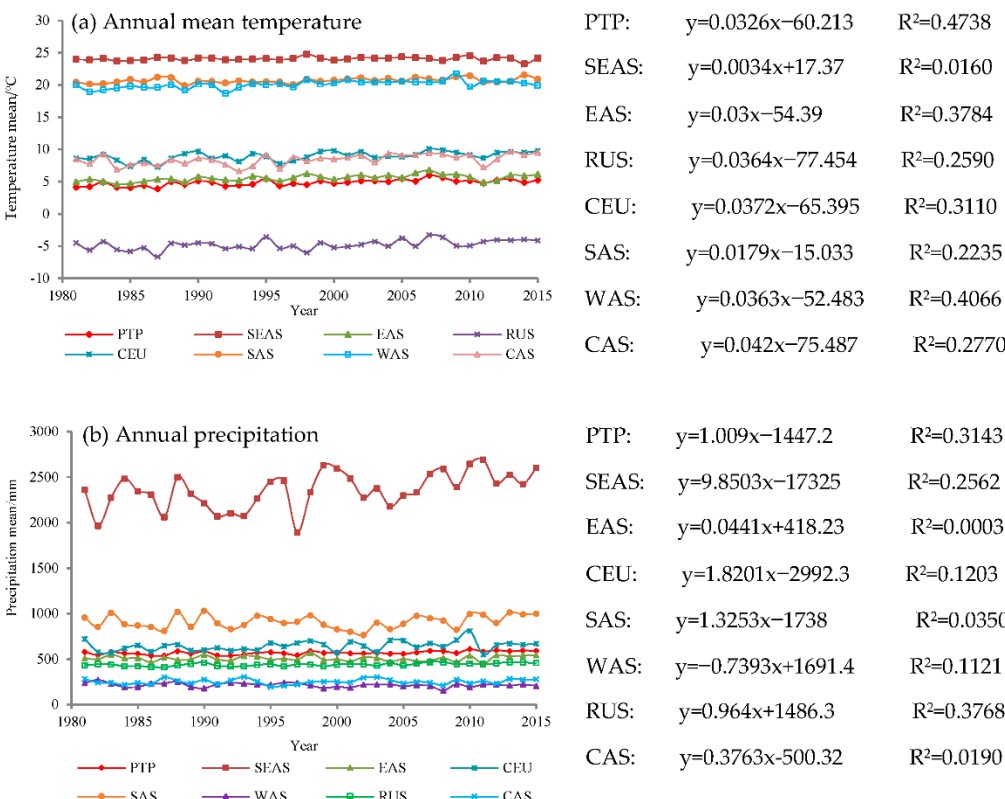

**Figure 12.** Inter-annual variations in annual mean temperature in subregions (**a**); inter-annual variations in annual precipitation in subregions (**b**).

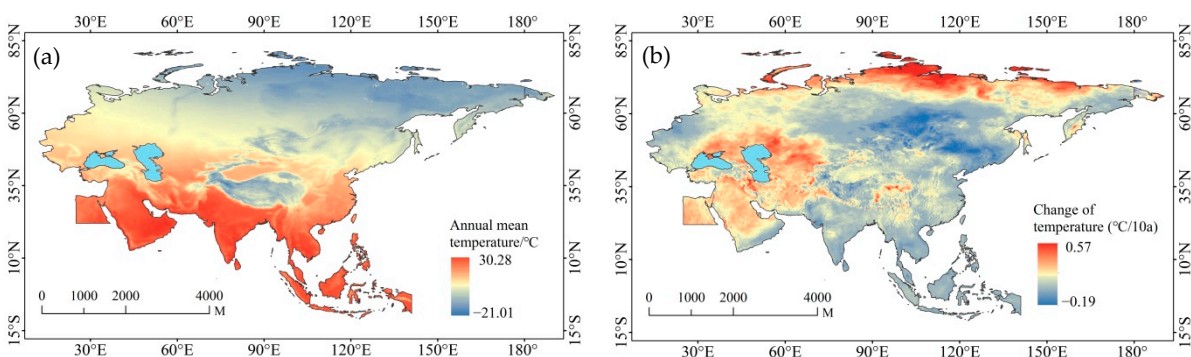

**Figure 13.** Distribution of annual mean temperature from 1982 to 2015 (**a**); change trend of annual mean temperature from 1982 to 2015 (**b**).

The spatial variability in the inter-annual rate of temperature change (*Slope*) was evident in the PTP region (Figure 13b), with values ranging from −0.19 to 0.57 °C•10a$^{-1}$ and a mean value of 0.08 °C•10a$^{-1}$. The zonal statistics in Figure 13b show that the inter-annual rate of temperature change (*Slope*) was as follows: Central Asia (0.14 °C•10a$^{-1}$) > West Asia (0.13 °C•10a$^{-1}$) > Russia (0.08 °C•a$^{-1}$) > Central and Eastern Europe (excluding Russia) (0.07 °C•a$^{-1}$) > South Asia (0.06 °C•a$^{-1}$) > East Asia (0.04 °C•a$^{-1}$) > Southeast Asia (0.02 °C•a$^{-1}$), showing that high latitudes warmed more rapidly than low latitudes.

From 1982 to 2015, the average annual precipitation in the PTP region was 567.72 mm, with an average trend rate of 10.09 mm•10a$^{-1}$ and showing a steady increasing trend (Figure 12b). The driest year was 1992, with an annual precipitation of 536.23 mm, which was 5.51% below the average precipitation; the year with the heaviest precipitation was 2010, which was 7.13% above the average precipitation. Based on the annual mean precipitation (Figure 14a) and the significance of the changing trend (Figure 14b), the annual

precipitation ranged from 15.23 to 8253.12 mm. The average annual precipitation for the subregions was ranked as follows: Southeast Asia (2355.86 mm) > South Asia (910.06 mm) > Central and Eastern Europe (excluding Russia) (644.35 mm) > East Asia (506.45 mm) > Russia (439.81 mm) > Central Asia (251.47 mm) > West Asia (214.37 mm). Indonesia had the highest annual precipitation with a multi-year average of 5000 mm. The Arabian Peninsula, Iran, Russia, and northern China had low precipitation. Turkey and Georgia are affected by the Mediterranean climate and, thus, had higher annual precipitation than the surrounding regions.

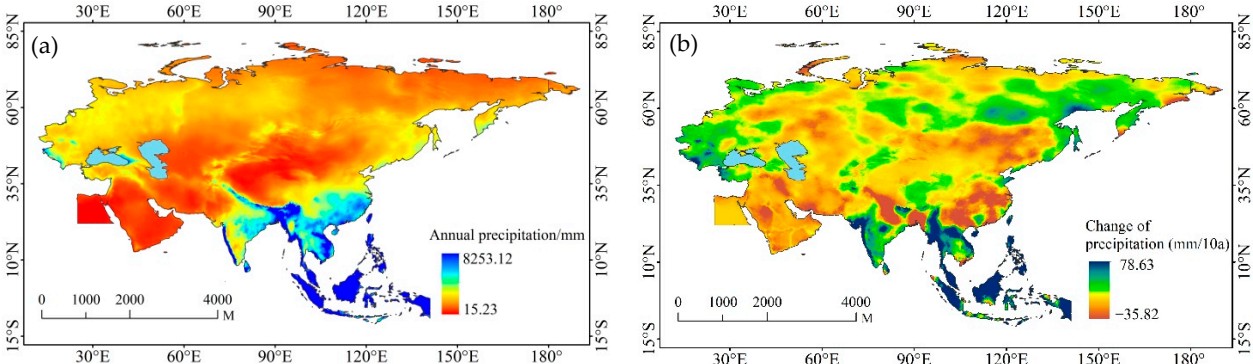

**Figure 14.** Distribution of annual precipitation from 1982 to 2015 (**a**); change trend of annual precipitation from 1982 to 2015 (**b**).

Regional differences in the inter-annual precipitation change rate (*Slope*) in the PTP region were obvious (Figure 14b), with values ranging from $-35.82$ to $78.63 \bullet 10a^{-1}$, with a mean of 9.6 mm$\bullet 10a^{-1}$. Zonal statistics for Figure 14b shows that the inter-annual rate of precipitation changes (*Slope*) is as follows: Southeast Asia ($58.52 \bullet 10a^{-1}$) > Central and Eastern Europe (excluding Russia) ($28.21 \bullet 10a^{-1}$) > South Asia ($12.15 \bullet 10a^{-1}$) > Russia ($10.56 \bullet 10a^{-1}$) > Central Asia ($3.67 \bullet 10a^{-1}$) > East Asia ($-3.38 \bullet 10a^{-1}$) > West Asia ($-7.56 \bullet 10a^{-1}$), with precipitation decreasing in East and West Asia and increasing in the other five regions. Precipitation tends to increase at high and low latitudes and decrease at mid-latitudes.

### 3.3.2. Spatial Correlation between *FVC* and Hydrothermal Conditions

Figure 15a, b show the correlation coefficients and the level of significance between the *FVC* and temperature in the study area over 34 years. The correlation coefficients were mostly positive, ranging from $-0.78$ to $0.81$, with an average of 0.13. In the seven subregions, the correlation coefficients between *FVC* and temperature were ranked as follows: Russia (0.24) > Central and Eastern Europe (excluding Russia) (0.22) > Central Asia (0.07) > East Asia (0.05) > Southeast Asia (0.02) > West Asia (0.01) > South Asia ($-0.13$), showing that the *FVC* was significantly correlated with temperature (excluding South Asia). The proportion of the *FVC* positively correlated with temperature was 70.30%, which was mostly found in high latitudes, such as Asia and Europe, the Siberian plains, and the middle and lower reaches of the Yangtze River in China. The proportion of the *FVC* that was negatively correlated with temperature was 29.6%, mainly found in the middle and low latitudes and near the equator. The proportion of significant positive, weak and significant negative correlations between *FVC* and air temperature were 20.41%, 75.43%, and 4.16%, respectively. In regions with a significant positive correlation, 28.73% of the region was covered by sparse shrubs, 26.69% by mixed forests, 9.27% by grasslands, 7.19% by cropland, and 28.12% by other land uses. In regions with a significant negative correlation, 37.19% was occupied by barren land, 28.92% by cropland, 14.51% by grasslands, 7.55% by sparse shrubs, and 11.83% by other land uses. In regions with weak correlations, grassland occupied 19.01% of the area, sparse shrubs covered 18.22%, agriculture covered 10.2%,

barren land covered 14.93%, croplands covered 11.98%, mixed forests covered 10.38%, and 25.48% was covered by other land uses.

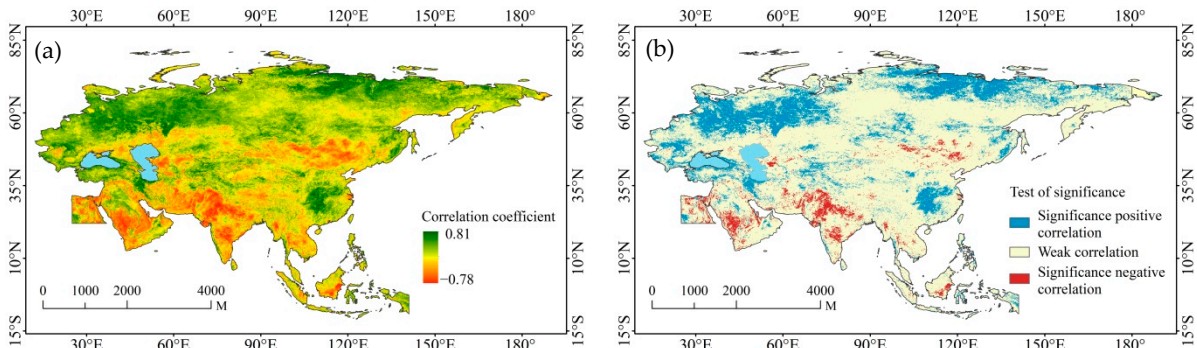

**Figure 15.** Correlation coefficient between *FVC* and temperature in the Pan-Third Pole region (**a**); test of significance between *FVC* and temperature in the Pan-Third Pole region (**b**).

The correlation coefficients and significance levels between *FVC* and precipitation during the 34-year period are shown in Figure 16a, b. The regional correlation coefficients between *FVC* and precipitation were mainly positive, ranging from −0.76 to 0.85, with a mean value of 0.05. Among the seven subregions, the correlation between *FVC* and precipitation was ranked as follows: Central Asia (0.31) > Central and Eastern Europe (excluding Russia) (0.23) > South Asia (0.12) > West Asia (0.09) > East Asia (0.06) > Southeast Asia (0.05) > Russia (−0.04). Most regions in Russia are influenced by polar atmospheric circulation and have an extremely cold climate, with heat being the predominant factor preventing plant growth. Precipitation was positively correlated with 54.5% of the *FVC*, mostly in Central Asia and the northern Black Sea, where vegetation cover is sparse and moisture is the most significant factor restricting plant growth. Precipitation was negatively correlated with 45.5% of the *FVC* in southern China, the high-latitude Siberian plains, Asia, and Europe. The results of the correlation between the *FVC* and precipitation indicated that the *FVC* and precipitation were significantly positively correlated in 13.1%, weakly correlated in 82.8%, and significantly negatively correlated in 4.1% of the study area, respectively. Of the regions with a significant positive correlation, 41.77% were grasslands, 19.95% were croplands, 9.67% were sparse shrubs, and 13.32% were other land-use types. In regions with a significant negative correlation, the proportion of mixed forest was 44.03%, the proportion of sparse shrubs was 10.56%, the proportion of evergreen needle-leaf forests was 8.42%, the proportion of croplands and natural vegetation mosaics was 6.69%, the proportion of evergreen broad-leaf forests was 5.92%, and 24.38% was covered by other land-use types. In regions with a weak correlation, the proportion of sparse shrubs was 21.95%, the proportion of grasslands was 13.37%, the proportion of barren land was 13.71%, the proportion of mixed forests was 13.63%, the proportion of cropland was 10.68%, and 26.66% was covered by other land-use types.

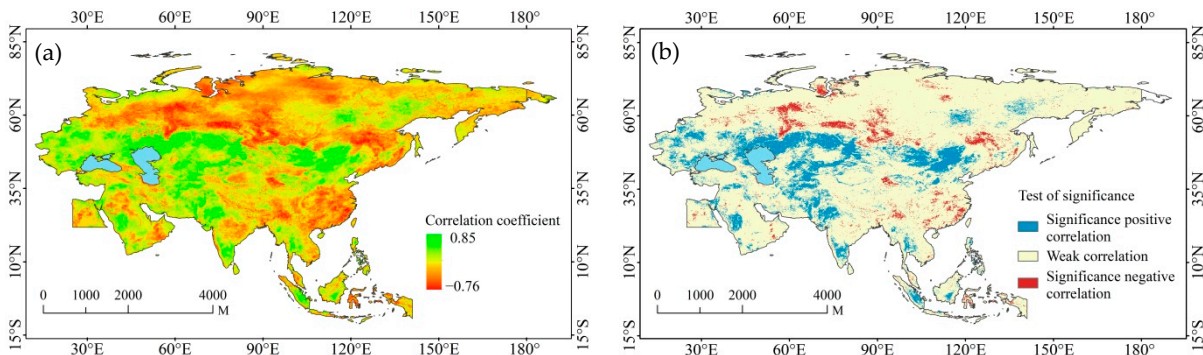

**Figure 16.** Correlation coefficient between *FVC* and precipitation in the Pan-Third Pole region (**a**); test of significance between *FVC* and precipitation in the Pan-Third Pole region (**b**).

## 4. Discussion

### 4.1. Topographic Differentiation Effect on FVC

The PTP region is typical of ecological regions in which topography, climate, and human activities interact. Topography (elevation, aspect, and slope) is one of the most significant non-zonal factors influencing vegetation distribution. It affects vegetation growth and geographical distribution by regulating light, heat, precipitation, and soil conditions [65,66]. Furthermore, elevation is the most important explanatory variable in topographic effects, which not only affects the distribution of temperature and moisture but also constrain human activities, resulting in vertical zonal changes in the distribution of vegetation. In the PTP region, the correlation coefficient between *FVC* and elevation was −0.31, with a negative correlation [24,65]. The *FVC* varies with the elevation gradient (Figure 17). Below 1400 m, the value of *FVC* increases with elevation; between 1400 and 2600 m, the value of *FVC* is flat and then slowly increases with elevation; and above 3500 m, the value of *FVC* is the lowest, close to 20%. The vegetation degradation type tends to decrease as elevation increases, the unchanged type increases gradually, and the vegetation improvement type first increases and then decreases (Figure 18). The trend of *FVC* (*Slope*) increases with elevation and, subsequently, decreases, with a maximum value of $0.039 \bullet 10a^{-1}$ for elevations between 1600 and 1800, followed by a value of $0.037 \bullet 10a^{-1}$ for elevations between 1200 and 1400, and a minimum value of $0.0015 \bullet 10a^{-1}$ for elevations more than 3500 m. With the change in elevation, the difference in vegetation cover is greatly associated with the natural environment and human activities. The temperature was colder at the higher elevations, where the vegetation growth was primarily limited by the lower temperature and not easily disturbed by human activities, with the strongest stability and the smallest proportion of vegetation improvement [67]. Regions with an elevation of less than 600 m are suitable for vegetation growth in terms of temperature and moisture, but they are also most vulnerable to human activities, and the stability of vegetation is low, but the vegetation cover has a greening trend in general; elevations between 1200 and 2000 m have a better ecological environment, with less human activity and the most obvious trend of vegetation greening. Therefore, we should be cautious of the browning trend in low elevation areas due to human activities in the PTP region.

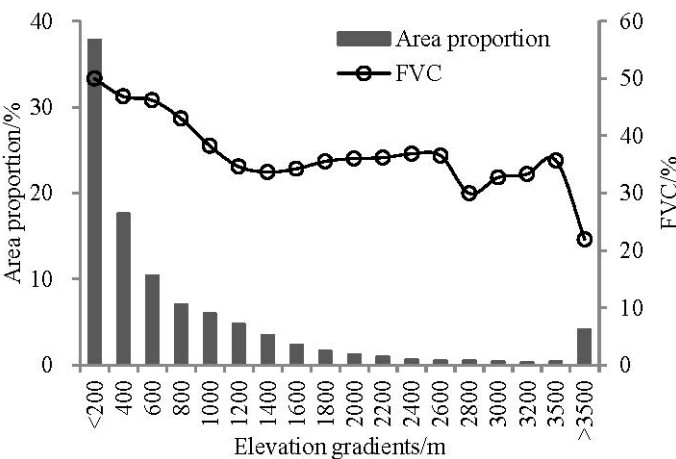

**Figure 17.** *FVC* change with elevation in PTP region.

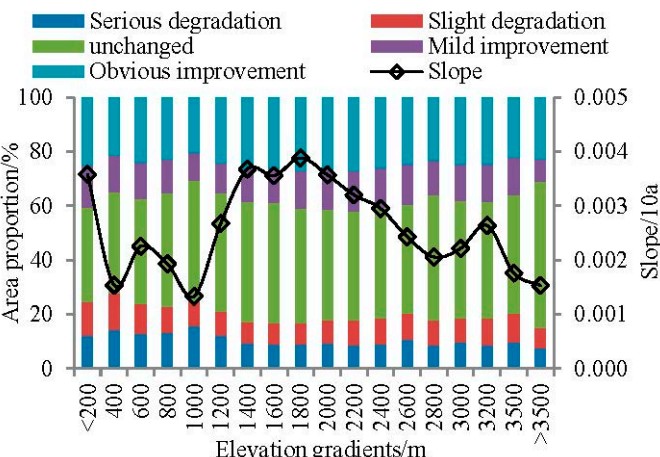

**Figure 18.** Area proportion of *FVC* change types at different elevation gradients in PTP region.

Changes in *FVC* from 1982 to 2015 were complex and spatially variable, with a slowly greening trend. Except for West Asia and South Asia, where the *FVC* decreased, the *FVC* increased in the other five subregions. Land use in West Asia and South Asia consists primarily of sparse grassland and bare land, with sparse surface vegetation. Grassland is significantly influenced by precipitation and human activities (over-cultivation, over-grazing, return to grass, etc.), making West Asia and South Asia vulnerable to global change. The other five subregions had modest increases in the *FVC*, whereas Central and Eastern Europe (excluding Russia) and Russia experienced the most rapid growth. The predominant land-use types in these regions are mixed forests, forested grasslands, sparse shrubs, farmland, and natural vegetation. Temperature was the primary controlling factor for vegetation growth in these two regions and the increase in temperature in the PTP region resulted in varying degrees of growth of temperate forests at high latitudes [68]. Usually, the SD is used to assess the variability of the long-term *FVC*. The geographic variability in SD is related to land-use type (Figure 5), and the land-use type in the low fluctuation region is mostly barren land, which is consistent with the fact that it will not change in a short period [61,62,69]. The majority of land-use types in low-fluctuation regions were forests, which tend to gradually increase the *FVC* in the absence of human activities. The *FVC* in grassland, scrub, and tundra, on the other hand, was affected by temperature, precipitation, and human activities and varies from year to year, showing moderate fluctuations. Wetland-water and urban land have a greater correlation with climate and human activities and their ecosystems are fragile and highly fluctuant.

*4.2. Spatial and Temporal Evolution Characteristics of the FVC under the Impact of Climate Change and Human Activities*

*FVC* is influenced by the climate, solar radiation, $CO_2$ concentration, human activities, topography, etc., with temperature and precipitation playing the most critical roles [70]. Climate change is an internal factor driving vegetation change, whereas human activities is an external factor, variations in land use induced by human activities, such as agriculture and urbanization, are important factors affecting the spatial patterns of vegetation [23], and the significance of human activities on vegetation change can no longer be underestimated [21,24]. The sixth IPCC report made it evident that human activities have been warming the atmosphere, oceans, and land for more than a century. As the signal of human influence has intensified, vegetation cover changes have strongly recorded the imprint of human activities, and the driving impacts of human construction and destruction activities on vegetation succession have become more significant [71,72]. Consequently, vegetation cover tends to be high in the east and low in the west, high in the south and low in the north, and decreases from east to west in the PTP region.

This study showed that the vegetation in the PTP region had a greening trend from 1982 to 2015 but with a high degree of regional variability; the main driving factors of the *FVC* increase in 34.71% of the study area were the combined effects of climate change and human activities. On the one hand, the surface temperature in the PTP region is on the rise, which prolongs the plant growth cycle and accelerates the decomposition of soil organic matter and the release of nutrients, thus, promoting the growth of vegetation. On the other hand, human activities could effectively increase vegetation cover at the local or even regional scale by improving agricultural management (e.g., fertilization and irrigation) and implementing vegetation construction projects.

Under the influence of climate change, the increasing and decreasing trends of $FVC_{CC}$ co-existed in the PTP region from 1982 to 2015. The $FVC_{CC}$ in Central and Eastern Europe (excluding Russia) increased the most significantly, followed by Russia. Warmer temperatures could enhance the activity of photosynthetic enzymes, slow the speed of chlorophyll degradation, and promote vegetation growth [73], especially in the northern high latitudes [41]. In contrast, the $FVC_{CC}$ of Southeast Asia is decreasing, and the land use types are mainly evergreen broad-leaf forests and evergreen needle-leaf forests. The sustained increase in temperature can lead to "physiological drought" of plants, which inhibits photosynthesis and growth rate, and may cause browning of vegetation, which is consistent with the found of Liu et al. (2022) that tropical vegetation tended to undergo browning in recent years [74]. Under the influence of human activities, the $FVC_{HA}$ in the PTP region showed a co-existing trend of increasing and decreasing from 1982 to 2015. $FVC_{HA}$ in Central and Eastern Europe (excluding Russia), East Asia and Central Asia is decreasing. In Central Asia, the $FVC_{HA}$ has shown an obvious decreasing trend due to rapid growth and population explosion, industrialization, urbanization, and the continuous degradation of the Aral Sea, an increasingly serious degree of desertification and a deteriorating ecological environment. Meanwhile, considering that the Central Asia countries experienced large changes in land-use followed by socio-economic disturbance after the Union of Soviet Socialist Republics (USSR) collapse (e.g., wars, revolutions, policy changes, and economic crises), these socio-economic factors are also likely to have contributed to the browning trend [39,75–78]. The vegetation in Inner Mongolia has been damaged by inappropriate mining, overgrazing, and other construction activities [43]. Hilker et al. (2014) showed that overgrazing is the single most important reason for desertification of the Mongolian Steppe [44]. Through economic expansion and urban construction, the *FVC* in Central and Eastern Europe (excluding Russia) has also shown a trend of degradation. In general, climate change influenced the increase in vegetation cover in the PTP region, but human activities positively influenced the growth of vegetation in the study area. Both climate change and human activities impacted the spatial-temporal evolution of the *FVC* in the PTP region.

### 4.3. Response of the FVC to Changes in Hydrothermal Conditions

The annual mean temperature and annual precipitation in the study area from 1982 to 2015 showed a fluctuating increasing trend and the climatic background was warm and humid, which provided good conditions for the growth of vegetation. The influence of temperature was generally greater than that of precipitation, which provided good conditions for vegetation growth, similar to the findings of previous studies [46,48]. The spatial pattern of the correlation between the *FVC* and temperature, precipitation had obvious spatial diversity and its spatial pattern was consistent with studies of global vegetation change [46,79,80]. The *FVC* was positively correlated with temperature in the temperate continental climate of the Asian and Siberian plains which have long and cold winters and short and warm summers but are affected by polar air masses year-round, resulting in an extremely frigid climate. The increase in temperature primarily promotes the growth of vegetation at high latitudes in the Northern Hemisphere in three ways: (1) the increase in temperature prolongs the growth period of plants in the Northern Hemisphere, consequently, enhancing the production of vegetation; (2) the increase in temperature improves nutrient effectiveness by promoting biogeochemical feedback; and (3) warming increases photosynthesis efficiency and water utilization [81]. For regions with high temperatures at low latitudes, near the equator, and other high-temperature water-scarce regions, an increase in temperature may increase the evaporation of soil water, resulting in a decrease in soil water moisture, which causes "physiological drought" of vegetation and suppresses photosynthesis and growth speed of plants. In these regions, only an increase in precipitation may promote plant growth [82], demonstrating an obvious negative correlation between temperature and plant growth. The lower elevation of the middle and lower sections of the Yangtze River Plain, as well as its relatively humid climate and sufficient precipitation, are suitable for plant growth. However, excessive precipitation will lead to an increase in cloudiness and a decrease in solar radiation, while the increase in soil moisture causes a relative increase in the evaporation of surface latent heat, all of which lower the temperature and decreases the photosynthetic rate, thus, suppressing the growth of vegetation. Consequently, plant growth in this region is more sensitive to temperature. The *FVC* is positively correlated with precipitation in arid and semi-arid regions, such as Central Asia and the northern Black Sea, where the annual precipitation is less than 300 mm. Vegetation change is more influenced by moisture than by other factors and an increase in precipitation substantially enhances vegetation growth [79]. In contrast, forests and shrubs in eastern China have dense vegetation and stable ecosystems that are not susceptible to changes in precipitation. The high-latitude plains of Siberia, Asia, and Europe have low temperatures year-round and heavy precipitation leads to an increase in cloudiness and a reduction in solar radiation, which lowers the temperature and is detrimental to plant growth. From the entire study area, the area of negative correlation between the *FVC* and temperature coincided with the area of positive correlation between the *FVC* and precipitation.

Significance testing of the FVC with temperature and precipitation showed that the proportion of the *FVC* with a significant positive association with temperature was greater than that with precipitation alone. Based on the correlation coefficient and significance of the *FVC* with temperature and precipitation, it is reasonable to conclude that temperature is the primary controlling factor of *FVC* change at high latitudes in the PTP region, which is consistent with the findings of Tucker et al. [34], Zhou et al. [83], and Piao et al. [84]. However, the response of vegetation to climate change is a nonlinear process with a compound impact, in which an increase in temperature enhances photosynthesis until the plant reaches the optimal temperature for photosynthesis. When the optimal temperature is exceeded, the temperature increase promotes crop respiration and accelerated nutrient consumption but also increases evapotranspiration and reduces organic matter accumulation. Moreover, maximum and minimum temperatures affect vegetative activities in a variety of ways. Changes in water may have some effect on vegetation activities; however, an increase in water can reduce vegetative activities by increasing cloudiness and humidity. Precipitation

is the most influential factor for plant growth in arid and semi-arid regions [14,85–87]. The threshold of hydrothermal conditions on vegetation change and the effect of non-climatic factors on vegetation change require further in-depth study in future research.

### 4.4. Limitations

This study provided a preliminary understanding of the spatial and temporal variability characteristics of *FVC* and its response to climate change and human activities in the PTP region and a basis for scientific assessment and decision support of regional ecological civilization-related resources and environmental issues that must be addressed urgently in the construction of the Green Silk Road. However, there were some limitations in this study: (1) Semi-monthly *FVC* was used to calculate the PTP region trend of vegetation cover. Because of the influence of snow cover, atmospheric aerosols, dust, and clouds, the semi-monthly *FVC* values often deviate from their nominal values, and the *FVC* trend may also be affected. Some other disturbances, such as forest fire, flood, soil moisture, overgrazing, drought, human activities, etc., also affected the trend of the *FVC*. However, these factors were not taken into account in this study, and future research will focus on them. (2) The spatial resolutions of the obtained NDVI, temperature, and precipitation data were inconsistent, and the analysis process was resampled to 8 km. There may be an effect on the accuracy of the results with mixed pixels in the data resampling process for monitoring *FVC* changes in the study area. However, in the future, more accurate change monitoring can be obtained using higher spatial resolution data to obtain more robust results. (3) When analyzing the association between the *FVC* and land-use types, only one-period land-use data were used and the impact of land-use type changes on *FVC* was not fully taken into account. (4) Partial correlation analysis and multiple regression residual analysis were used to analyze the relationships between vegetation cover, climatic factors, and human activities. Although the driving factors of vegetation cover have been identified, geographic detectors and machine learning methods for the quantitative analysis of the driving factors are a future research focus. To predict vegetation change trends, which are crucial for the advancement of vegetation restoration, future studies should consider more influencing factors and screen out those with greater contribution rates. (5) In this study, only NDVI was used to calculate *FVC*, which is insufficient. Some other formulations of *FVC* should be taken into consideration in future studies. For example, Jiang et al. (2006) found that scaled difference vegetation index (SDVI), a scale-invariant index based on linear spectral mixing of red and near-infrared reflectance, is a more suitable and robust approach for retrieval of vegetation fraction with remote sensing data, particularly over heterogeneous surfaces [88]. Furthermore, the various methods should be compared to choose the optimal solution to calculate the *FVC*.

### 5. Conclusions

The spatio-temporal succession characteristics of the *FVC* in the PTP region and the correlation coefficient between the *FVC* and climate factors and human activities were analyzed based on the GIMMS NDVI3g dataset from 1982 to 2015, combined with precipitation and temperature data for the same period. The following main conclusions were drawn:

(1) The average *FVC* over the past 34 years in the PTP region was 45.65%, with significant regional differences in macroscopic patterns, with Southeast Asia having the highest average *FVC* at 94.16% and West Asia the lowest at 17.08%.

(2) During the 34 years, the slope of the *FVC* change in the PTP region fluctuated between $-0.15 \bullet 10a^{-1}$ and $0.36 \bullet 10a^{-1}$, with an average slope of $0.003 \bullet 10a^{-1}$. The *FVC* change may be roughly divided into three phases: the fast-rising phase from 1982 to 1994; the browning phase from 1994 to 2003; and the steady greening phase after 2003. The proportions of areas with obvious improvement, mild improvement, unchanged, slight degradation, and serious degradation in the *FVC* were 23.83%, 13.53%, 39.29%, 11.16%, and 12.19%, respectively. In subregions, the trend of change

(*Slope*) in the *FVC* was greater than zero, with the greatest increasing trend in South Asia (*Slope* = $0.0078 \bullet 10a^{-1}$) and the lowest in Central Asia (*Slope* = $0.0002 \bullet 10a^{-1}$). Overall, vegetation in the Pan-Third Pole region showed a greening trend over the 34-year period.

(3) The effects of climate change and human activities on the *FVC* in the PTP region were spatially heterogeneous but were mainly positive. In the PTP region, the impacts of climate change and human activities on the average growing season *FVC* changes from 1982 to 2015 were $0.0013 \bullet 10a^{-1}$ and $0.0011 \bullet 10a^{-1}$, respectively. Climate change and human activities were the driving factors of the *FVC* increases in South Asia, West Asia, and Russia; climate change was the driving factor of the *FVC* increase in Central and Eastern Europe (excluding Russia), East Asia, and Central Asia; and human activities were the driving factors of the *FVC* increase in Southeast Asia.

(4) From 1982 to 2015, the climate of the PTP region tended to be warm and humid, with 70.3% of the *FVC* positively correlated with temperature and 54.5% of the *FVC* positively correlated with precipitation. In the growing season, the *FVC* was positively correlated with the annual mean temperature at high latitudes, while for arid and semi-arid regions in the low and middle latitudes, the *FVC* during the growing season was negatively correlated with temperature and positively correlated with precipitation.

**Author Contributions:** Conceptualization, X.Y. and Q.Y.; methodology, X.Y. and M.Y.; software, X.Y.; investigation, M.Y.; resources, X.Y. and M.Y.; data curation, X.Y. and Q.Y.; writing—original draft preparation, X.Y.; writing—review and editing, Q.Y.; project administration, Q.Y.; research group leader, Q.Y. All authors have read and agreed to the published version of the manuscript.

**Funding:** This research was supported by the Strategic Priority Research Program of Chinese Academy of Sciences, Grant No. XDA20040202.

**Data Availability Statement:** Not applicable.

**Conflicts of Interest:** The authors declare no conflict of interest.

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
