# Peer review of "Spatio-Temporal Patterns and Driving Factors of Vegetation Change in the Pan-Third Pole Region"

_remotesensing, doi:10.3390/rs14174402_

Round 1
Reviewer 1 Report
The manuscript focused on spatiotemporal patterns and driving factors of vegetation in the pan-third pole region.
Comments,
- I recommend updating the title to be " ..... driving forces of vegetation change ...."
- What do you mean by fractional vegetation cover? if it is fractional then it will need a higher resolution remote sensing data to be detected, but you selected MODIS to have a high coverage but still, you can not call it fractional vegetation cover.
- Why did you select this time period 1982-2015, provide an explanation and/or clarification, and explain how did you get MODIS data for 1982 to 2000?
- It will be better if you can include a chart in the materials and methods section to show the methodology adopted in this study.
Reviewer 2 Report
Dear Authors,
I think that your manuscript entitled “Spatio‐temporal Patterns and Driving Factors of Vegetation in the Pan‐Third Pole Region” brings quite interesting results to the impact of global warming on changes in vegetation cover. This work is important from a scientific and practical point of view and could be suitable for publication in the Remote Sensing journal. However, the manuscript should be improved before publication.
First of all, the approach the authors use, namely the spatio-temporal variation of fractal vegetation coverage using linear regression analysis, standard deviation, and correlation coefficient to explore its response mechanism to climate change and human activities is approximate, because of the big-scale investigations. Of course, one can imagine the need for such investigations for some global analyses, but one ought to be aware of their limitations. For instance, the results shown in Fig 2a), 3, and 9 (inter‐annual variations and regression) are averaged over various more local climatic and geological conditions. Their interpretation is difficult, if possible. Besides, the limitations arise not only from the scale of the investigations but also from the oversimplified classical statistical approach, which does not take into consideration spatial statistics or geostatistics combined with the remote sensing observations of the vegetation ecosystems. Besides, when writing about the vegetation cover the authors neglect important factors such as soil moisture, overgrazing, and dust influence on remote sensing observations, etc. Subsection 4.4 should be slightly supplemented by the mentions about these factors. Another important point is that the literature review is scarce not to say done rather carelessly as for a such big and varied area of the investigation. References from the EAS (mainly from China) dominate, while the remaining sub-regions are practically omitted from the bibliography. It concerns both vegetation cover and other environmental issues as rainfall, soil moisture, etc. This should be supplemented and improved. I would suggest to take into account, for example, the following items, among others chosen by the authors.
Hao H, Chen Y, Xu J, Li Z, Li Y, Kayumba PM. Water Deficit May Cause Vegetation Browning in Central Asia. Remote Sensing. 2022; 14(11):2574. https://doi.org/10.3390/rs14112574
J. Zawadzki , C.J. Cieszewski, M. Zasada and R.C. Lowe , Applying geostatistics for investigations of forest ecosystems using remote sensing imagery. Silva Fenica. 2005. 39(4):599-617.
Rustanto, A.; Booij, M.J. Evaluation of MODIS-Landsat and AVHRR-Landsat NDVI data fusion using a single pair base reference image: A case study in a tropical upstream catchment on Java, Indonesia. Int. J. Digit. Earth 2022, 15, 164–197.
Dass, P., Rawlins, M. A., Kimball, J. S. and Kim, Y.: Environmental controls on the greening of terrestrial vegetation across northern Eurasia, Biogeosciences Discuss, 12(12), 9121–9162, doi:10.5194/bgd-12-9121-2015, 2015.
Roy, P. S., Behera, M. D., Murthy, M. S. R., Roy, A., Singh, S., Kushwaha, S. P. S., … & Gupta, S. (2015). New vegetation type map of India prepared using satellite remote sensing: Comparison with global vegetation maps and utilities. International Journal of Applied Earth Observation and Geoinformation, 39, 142-159.
Namdari, S.; Zghair Alnasrawi, A.I.; Ghorbanzadeh, O.; Sorooshian, A.; Kamran, K.V.; Ghamisi, P. Time Series of Remote Sensing Data for Interaction Analysis of the Vegetation Coverage and Dust Activity in the Middle East. Remote Sens. 2022, 14, 2963. 10.3390/rs14132963
Hilker, T., Natsagdorj, E., Waring, R. H., Lyapustin, A., & Wang, Y. (2014). Satellite observed widespread decline in Mongolian grasslands largely due to overgrazing. Global Change Biology, 20(2), 418-428. doi:10.1111/gcb.12365
etc.
Summary
This is a systematic and good manuscript, having interesting results, but using a somewhat oversimplified approach. The introduction and bibliography have to be supplemented with references from outside the EAS region and China, as well as some methodological ones.
I recommend it for publication after major revision. Sincerely yours
Reviewer 3 Report
see attached file

Round 2
Reviewer 1 Report
Accept in present form.
Author Response
Thanks to the reviewer for the recognition.
Reviewer 2 Report
Dear Authors
Thank you for the thorough revision of your manuscript, and your explanations. They satisfy me.
Sincerely Yours,
Reviewer
Author Response

(The authors gave the same response as above.)
